# Migration of $^{238}$U and $^{226}$Ra Radionuclides in Technogenic Permafrost Taiga Landscapes of Southern Yakutia, Russia

Aleksandr Chevychelov *, Petr Sobakin, Aleksey Gorokhov, Lubov Kuznetsova and Aleksey Alekseev 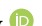

Institute for Biological Problems of Cryolithozone SB RAS, 677980 Yakutsk, Russia; radioecolog@yandex.ru (P.S.); algor64@mail.ru (A.G.); likkol@yandex.ru (L.K.); alex3.fromru@gmail.com (A.A.)
* Correspondence: chev.soil@list.ru

**Abstract:** This article describes the features and migration patterns of natural long-lived heavy radionuclides $^{238}$U and $^{226}$Ra in the major components of the environment including rocks, river waters, soils, and vegetation of permafrost taiga landscapes of Southern Yakutia, which helped us to understand the scale and levels of their radioactive contamination. Different methods have been used in this study to determine the content of $^{238}$U and $^{226}$Ra in various samples, including gamma-ray spectrometry, X-ray spectroscopy, laser excited luminescence, and emanation method. It was determined that the main source of radioactive pollution of soil and vegetation cover, as well as surface waters in these technogenic landscapes, are the dumps of radioactive rock that were formed here as the result of geological exploration carried out in this area during the last third of the 20th century. The rocks studied were initially characterized by a coarse, mainly stony gravelly composition and contrasting radiation parameters, where the gamma radiation exposure rate varied between 1.71 and 16.7 μSv/h, and the contents of $^{238}$U and $^{226}$Ra were within the range 126–1620 mg/kg and 428–5508 × $10^{-7}$ mg/kg, respectively, and the $^{226}$Ra: $^{238}$U ratio was 1.0. This ratio shifted later on from the equilibrium state towards the excess of either $^{238}$U or $^{226}$Ra, due to the processes of air, water, and biogenic migration. Two types of $^{238}$U and $^{226}$Ra radionuclides migration were observed in studied soils, namely aerotechnogenic and hydrotechnogenic, each of which results in a different intraprofile radionuclide distribution and different levels of radioactive contamination. In this study, we also identified plants capable of selective accumulation of certain radionuclides, including Siberian mountain ash *(Sorbus sibiricus)*, which selectively absorbs $^{226}$Ra, and terrestrial green and aquatic mosses, which accumulate significant amounts of $^{238}$U.

**Keywords:** cryolithozone; technogenic landscapes; radionuclides $^{238}$U; $^{226}$Ra; migration; content; distribution

## 1. Introduction

The term technogenesis encompasses a number of geochemical processes associated with human activity that is accompanied by the extraction, concentration, and redistribution of chemical elements in the environment. The forecast of technogenic impact and the substantiation of rational methods for protecting the natural environment should be based on knowledge of migration patterns of chemical elements in various natural geochemical environments of the Earth's surface [1].

During the exploration and development of deposits with a high content of natural radionuclides, the extra amounts of natural radioactive elements enter different components of the environment. The nature and scale of such flow depends on the type of deposits, the technology of exploration and development of subsoil, the local geochemical features of the environment, and other factors [2–4].

Uranium is the heaviest known chemical element occurring in appropriate amounts in the Earth's crust. Its atomic number is 92 and its standard atomic weight is 238.07. There are three long-lived uranium isotopes found in nature: $^{238}$U, $^{235}$U, and $^{234}$U, which

are characterized by the following abundance: 99.27%, 0.72%, and 0.01%, and half-lives: $4.51 \times 10^9$, $7.13 \times 10^8$, and $2.47 \times 10^5$ years, respectively [5]. Uranium-238 is the most abundant uranium isotope found in rocks, weathering crusts, and soils, which makes up 99.3% of the total uranium on Earth. According to the geochemical classification of elements, uranium belongs to the group of lithophilic and siderophile elements with variable valence, capable of forming both cations and anions. Physicochemical migration plays a very important role in uranium geochemistry; it is an active migrant in geotherms and in hypergenesis zones, and it concentrates on barriers of many classes, therefore uranium belongs to the group of mobile and weakly mobile elements in oxidizing conditions, inert in reducing environments (gley and hydrogen sulfide), and deposits on hydrogen sulfide and gley barriers [6].

Radium is represented in nature by four isotopes: $^{228}$Ra, $^{226}$Ra, $^{224}$Ra, and $^{223}$Ra, each of which has a significantly different half-life and plays a different role in geochemical processes. The most studied and long-lived radium isotope ($^{238}$U series), $^{226}$Ra, has a half-life of $1.6 \times 10^3$ years. In rocks and minerals, $^{226}$Ra is usually present in equilibrium with $^{238}$U, with a weight ratio $^{226}$Ra: $^{238}$U = $3.4 \times 10^{-7}$; this ratio in activity units (Bq), also known as the coefficient of radioactive equilibrium, is equal to 1.0. The shift of this equilibrium is often observed in both directions, towards $^{226}$Ra and towards $^{238}$U. The specifics of $^{226}$Ra geochemistry are associated with the fact that radium is an alkaline earth metal (with chemical properties very close to those of barium), and also that it is related to $^{238}$U by its origin.

According to the specifics of hypergenic migration, radium belongs to the group of mobile elements with constant valence, for which the values of water migration coefficient Kx are equal to $n - 10 \times n$, where n is an order of magnitude. This means that radium actively migrates in waters of the hypergenesis zone. In aqueous solutions, radium can be in the ionic, molecular, or pseudo-colloidal form [6]. At the same time, for the soils of humid zones, the following migration series of elements Ra > U > Th is maintained [7]. However, in the zone of hypergenesis, depending on the physicochemical conditions of radionuclide migration, both the relative mobility of $^{226}$Ra and $^{238}$U and the values of the weight ($3.4 \times 10^{-7}$) and radioactive equilibrium (1.0) radium–uranium ratios can shift to either one or the other side [8].

Many researchers have studied the migration of natural $^{238}$U and $^{226}$Ra radionuclides in the major components of technogenic landscapes and investigated the radioecological situation at radiation hazardous objects in different natural zones [8–21]. Most of the research data, however, were collected in nonpermafrost regions of Russia and other countries. The results of the comprehensive work presented in this article were first obtained in the permafrost area in the territory of Southern Yakutia.

The goal of this study was to determine the content levels and distribution characteristics of $^{238}$U and $^{226}$Ra radionuclides in the main components of permafrost taiga landscapes (rocks, surface waters, soils, and vegetation), and to determine the scale of air and water migration of these radionuclides in the natural conditions of Southern Yakutia.

## 2. Materials and Methods

The studies were carried out in the Elkon uranium region in Southern Yakutia at different times, beginning from 2005 (Figure 1). This area is geomorphologically confined to the Elkon horst, which is a mountain uplift in the Aldan Highlands, characterized by low and medium mountain relief and the predominance of mountain taiga vegetation. The climate of the study area is continental, cold humid, and superhumid. In the soil cover of the region, podzols (Albic Podzols) and podburs (Entic Podzols) are formed in automorphic positions, and alluvial soils (Fluvisols) are formed in the floodplains of rivers and creeks.

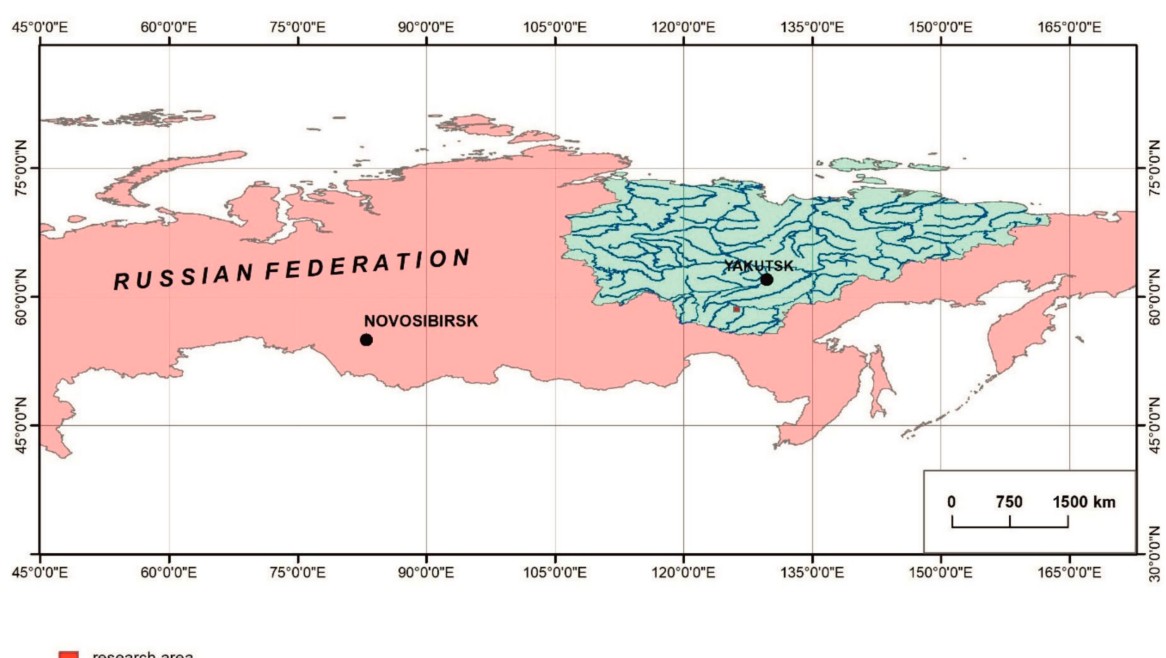

**Figure 1.** Location of the study area.

The natural mountain–taiga landscape of the Elkon horst was heavily disturbed by a long-term and large-scale geological prospecting for radioactive raw materials, which were carried out here during the last third of the 20th century. As a result of these works, more than 1 million tons of ore were extracted and stored on the surface as dumps. The total amount of uranium contained in this ore reaches 2000 tons. This fact assumes a high degree and significant scale of radionuclide contamination of the soil, vegetation cover, and surface waters of the area, considering the landscape and climatic conditions of this region. The dumps, as a rule, are confined to the bottoms and slopes of valleys of small rivers and creeks, and their constituent components, including natural radionuclides, subjected to intense wind and water dispersion by the processes of hypergenic weathering [22].

The major components of technogenic permafrost taiga landscapes of the study area, including rocks of radioactive dumps, surface waters, soils, and vegetation were the primary objects of this study. The first step was to carry out a gamma radiation survey using an SRP-68-01 radiometer at all selected sites and outside of their territories. Route and aerial surveys were conducted using an arbitrary observation network, according to the methodology adopted in geology and radiation ecology [23]. The gamma radiation exposure rate measurements were made at a distance of 1 m from the emitting surface, and also in close proximity within 0.1 m. The evaluation of gamma radiation exposure rates was done by calculation of the arithmetic mean of 3–5 measurements at each point of the survey. After the gamma survey, sites were selected for sampling rocks, surface waters, soils, and plants.

*Rocks of radioactive dumps.* Rock samples were represented by a mixture of fine earth (particles less than 1 mm), gravel (1–3 mm), and stones (>3 mm); the latter were dried, grounded, and sieved through a 1 mm mesh, and then analyzed.

*Surface waters.* Water samples were collected from rivers and creeks flowing through radioactively contaminated areas. Samples were taken at the base of radioactive dumps, and also at different distances upstream and downstream from the contamination source, generally during the period of summer–fall baseflow. The volume of water sample depended on downstream analysis, so 30 mL of water was sufficient for uranium detection, while radium detection required as much as 1 L of water. Moreover, additional 1 L water samples were collected for subsequent general chemical analysis of the surface waters. For the determination of chemical parameters of tested water samples, we used a number of well-known analytical methods described elsewhere [24].

*Soils.* Soil morphology was studied in soil pits that were dug near the dumps in the areas of radionuclide contamination, usually during late summer, by the time of maximum seasonal soil thawing. Soil samples were collected layer by layer every 1–4 cm throughout the soil profile considering the boundaries of the genetic horizons. Similar to the rock samples, soil samples were dried, ground, and analyzed. Generally accepted soil science methods, including comparative, analytical, geographical, geochemical, and others, were used in this work [25–27]. The chemical composition and properties of soils were determined according to the methods adopted in soil science [28].

*Plants.* Plant samples were collected by the end of the growing season, at close proximity to the shallow pits of fine earth and soil pits. Trees and shrubs were cut down at the base of the trunks and divided into leaves/needles, branches, and trunks. The aboveground parts of grasses, mosses, and lichens were cut off from a certain area. All plant samples were dried and ashed at 500 °C and then analyzed.

Different methods were used in this study to determine the content of natural $^{238}$U and $^{226}$Ra radionuclides in prepared samples of fine rock, soil, water, and plants, including gamma-ray spectrometry, X-ray spectroscopy, laser excited luminescence, emanation, and other methods widely used in geology and radioecology [9,29,30].

The $^{238}$U content in rock, soil, and ash samples in the condition of radioactive equilibrium was determined by gamma-ray spectrometry using the daughter $^{226}$Ra and Progress-Gamma multichannel analyzer with a 63 mm × 63 mm NaI(Tl) scintillation detector, with 7.3% resolution in γ- $^{137}$Cs lines (661.6 keV).

Gamma spectrometry measurements were carried out using 1 L Marinelli beakers. The γ-spectra were processed by the matrix method using the Progress software [31]; the measurement error did not exceed 10–15%. The content of uranium-238 was also determined by X-ray spectroscopy using an ARF-6M analyzer, with a sensitivity threshold for uranium-238 of 2 mg/kg and a determination error of ±10% [32]. The detection of uranium-238 in water samples was performed by the method of laser excited luminescence using an AUF-101-Angara fluorometer with the sensitivity for uranium-238 of $2 \times 10^{-5}$ mg/L, and the error, in this case, did not exceed ±15%.

Radium in solid samples, including rock, soil, and plant ash, was determined by the emanation method using the Alfa-1M device. The detection sensitivity was $2 \times 10^{-8}$ mg/kg, and the determination error did not exceed ±20%. When determining the content of radium in water samples, the sensitivity of this method was $2 \times 10^{-9}$ mg/L, and the error did not exceed ±15%.

The goal of this study was achieved by using both geochemical (radionuclide migration) and radioecological (radioactive contamination) approaches. Therefore, the content of $^{238}$U and $^{226}$Ra radionuclides with long half-lives ($4.5 \times 10^9$ and $1.6 \times 10^3$ years, respectively) is presented in this article as a percentage by weight and in radioactivity units, Bq/kg.

## 3. Results and Discussion

### 3.1. Dumps of Radioactive Rock

During geological exploration, dozens of uranium deposits and ore occurrences were found in the Elkon uranium region. All of them belong to a single gold–uranium formation, the uranium and gold of which are associated with potassium metasomatites of the activation zones of ancient faults in Central Aldan [33]. Brannerite is the major uranium-bearing mineral unevenly distributed in ore deposits [34,35]. As the scale of geological exploration works increased in this area, large volumes of radioactive rocks accumulated on the surface near mines and adits. These rocks were extracted from the bowels of the Earth to the hypergenesis zone on the surface and left as dumps with different volumes and radioactivity levels. The largest dumps, some of which are close to radioactive ores in terms of radiation levels, are located in the Yuzhnaya zone of the Elkon uranium region (Figure 2), and at the Kurung and Akin sites (Figure 3).

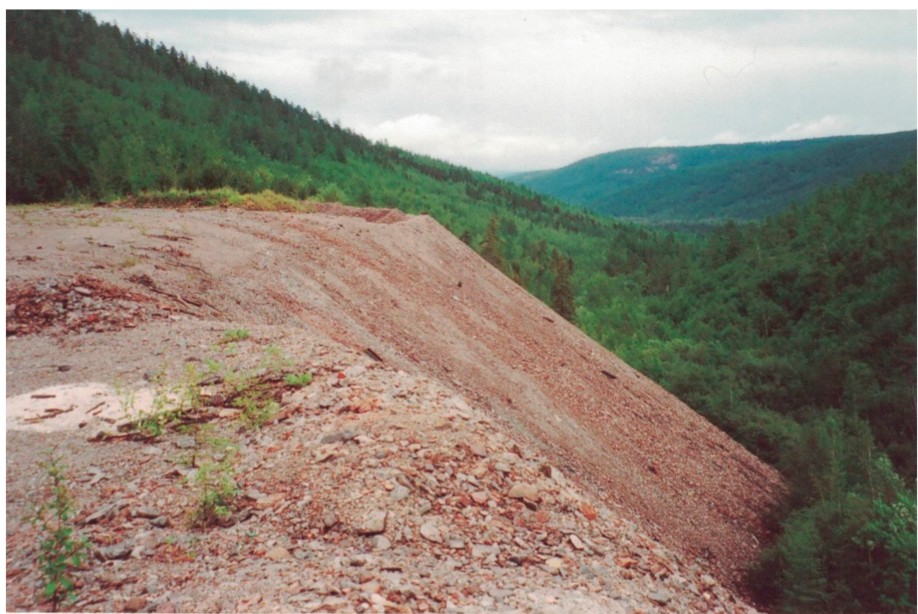

**Figure 2.** Dumps of radioactive rock at the Kurung site.

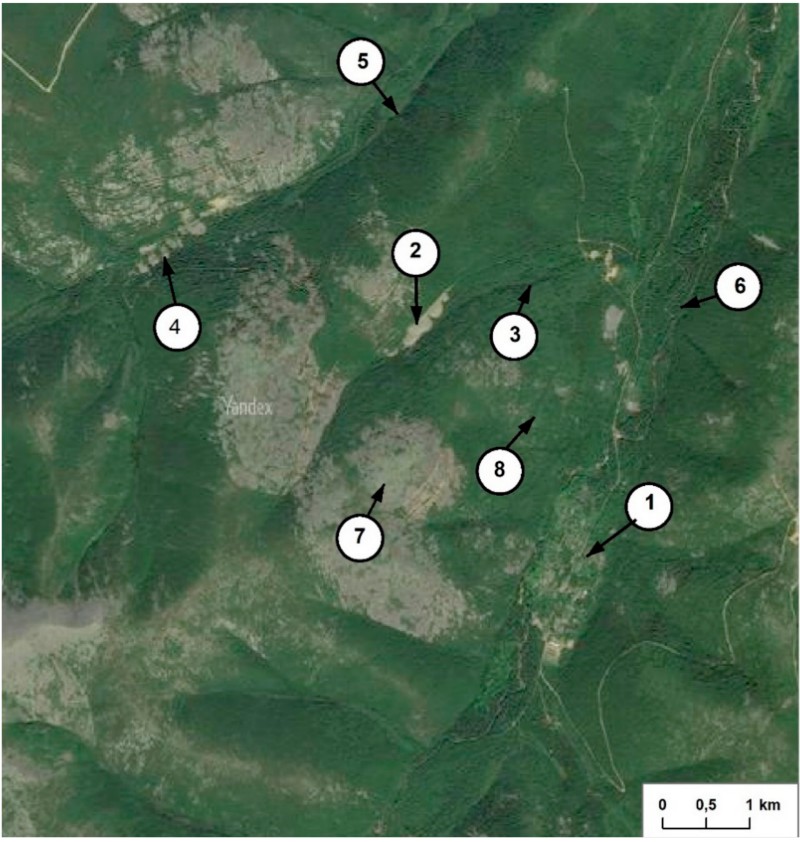

**Figure 3.** Satellite image of the study area: 1—Zarechny village; 2—Kurung site; 3—Propadayushiy Creek; 4—Akin site; 5—Akin Creek; 6—Kurung River; 7—tundra mountain tops and stony heaths; 8—watershed slopes with mountain–taiga vegetation.

The rock of the dumps is primarily represented by a stony gravelly mixture of coarse earth, with a size of particles of more than 1 mm in diameter. According to our data, 77–89% of dumped rock matter, by weight, is represented by coarse soil particles with a diameter of more than 1 mm, 10–16% of which is gravel (particle size 1–3 mm), 61–79% stones of various sizes (particles larger than 3 mm), and only 11–23% represented by fine

earth (particles smaller than 1 mm) [36]. Thus, the physical characteristics of the rocks at the Kurung site, including their radiological parameters, were completely determined by the properties of the gravel–rocky portion of these dumps, which is a geologically closed system in which uranium and radium are in a state of radioactive equilibrium, where $^{226}$Ra: $^{238}$U ratio is equal to 1.0.

We also observed significant variability in the exposure rate values, and in the content of $^{238}$U and $^{226}$Ra in the dump rock, which were within the range of 1.71–16.7 μSv/h, 126–1620 mg/kg, and 428–5508 × 10$^{-7}$ mg/kg, respectively (Table 1).

**Table 1.** Content of $^{238}$U and $^{226}$Ra in the gravel–stony portion of the dumps at the Kurung site.

| Dump No | Exposure Rate, μSv/h | $^{238}$U | | $^{226}$Ra | | $^{226}$Ra: $^{238}$U |
|---|---|---|---|---|---|---|
| | | Bq/kg | mg/kg | Bq/kg | n × 10$^{-7}$ mg/kg | |
| 1 | 16.7 | 19926 | 1620 | 19926 | 5508 | 1.0 |
| | 10.1 | 11870 | 965 | 11870 | 3281 | 1.0 |
| | 8.6 | 10049 | 817 | 10049 | 2961 | 1.0 |
| 2 | 7.26 | 8376 | 681 | 8376 | 2315 | 1.0 |
| | 4.74 | 5277 | 429 | 5277 | 1459 | 1.0 |
| | 2.23 | 2189 | 178 | 2189 | 605 | 1.0 |
| 3 | 3.91 | 4256 | 346 | 4256 | 1176 | 1.0 |
| | 2.62 | 3223 | 217 | 3223 | 738 | 1.0 |
| | 1.71 | 1550 | 126 | 1550 | 428 | 1.0 |

The rocks studied have been exposed for about 40–50 years and due to the climatic conditions of Southern Yakutia, they were subjected to intense physical and chemical weathering. As the result of these processes, we observed the increase in the content of fine earth particles, predominantly sandy (1.0–0.05 mm) and coarse-grained (0.05–0.01 mm) fractions. Under these conditions, the radioactive equilibrium in the fine earth fraction of the dumps is shifted towards $^{226}$Ra, and the $^{226}$Ra: $^{238}$U ratio varies from 1.10 to 1.19 (Table 2).

**Table 2.** Content of $^{238}$U and $^{226}$Ra in the fine earth fraction of the dumps of the Kurung site.

| Dump No | Exposure Rate, μSv/h | $^{238}$U | | $^{226}$Ra | | $^{226}$Ra: $^{238}$U |
|---|---|---|---|---|---|---|
| | | Bq/kg | mg/kg | Bq/kg | n × 10$^{-7}$ mg/kg | |
| 1 | 12 | 9397 | 882 | 10,849 | 2999 | 1.15 |
| 1 | 7 | 5313 | 432 | 5936 | 1641 | 1.12 |
| 2 | 4 | 3677 | 299 | 4044 | 1117 | 1.10 |
| 3 | 2.5 | 2664 | 179 | 3183 | 729 | 1.19 |

Therefore, the cold humid climate of Southern Yakutia and the predominance of mountain–taiga vegetation facilitate intense leaching of $^{238}$U from dump rocks as compared with $^{226}$Ra under conditions of acidic oxidative weathering. This feature was even more evident when analyzing the chemical composition of the surface waters of the study area.

### 3.2. Surface Waters

Little is known about the radioactive characteristics of surface waters of the permafrost zone. Here we conducted an elaborate study of the chemical composition of surface waters, including the determination of uranium and radium concentrations depending on the distance to radioactive dumps, using the example of Propadayushiy Creek, which drains rock dumps at the Kurung site (Figure 3). Water samples from Propadayushiy Creek were taken upstream and downstream from the radioactive dumps at fixed distances of 500 m from a contamination source. In addition, water samples from the Kurung, Elkon, and Aldan rivers were collected outside the zones of technogenic pollution [37].

According to the established classification of total water mineralization [38], the waters studied were characterized as ultrafresh waters, and in terms of the ion composition, waters of natural landscapes were characterized as hydrocarbonate–calcium, while waters of technogenic landscapes were defined as sulfate–calcium (Table 3).

**Table 3.** Ionic composition of surface waters of technogenic and natural landscapes of Southern Yakutia.

| # | Sampling Site | pH | Ions, mg/L/% eq | | | | | | | Total Ions, mg/L |
|---|---|---|---|---|---|---|---|---|---|---|
| | | | $Ca^{+2}$ | $Mg^{+2}$ | $Na^+$ | $K^+$ | $HCO_3^-$ | $SO_4^{-2}$ | $Cl^-$ | |
| | Technogenic landscapes | | | | | | | | | |
| 1 | Propadayushiy Creek, beginning of dumps | 7.7 | 5.8 / 22.3 | 1.9 / 12.3 | 4.0 / 13.4 | 1.0 / 2.0 | 12.8 / 16.2 | 20.7 / 33.1 | 0.3 / 0.7 | 46.5 |
| 2 | 500 m downstream the dumps | 6.9 | 9.4 / 23.7 | 3.9 / 16.2 | 4.0 / 8.8 | 1.0 / 1.3 | 15.3 / 12.6 | 35.1 / 36.9 | 0.3 / 0.5 | 69.0 |
| 3 | 1000 m downstream the dumps | 6.7 | 8.2 / 24.2 | 3.4 / 16.6 | 3.0 / 7.7 | 1.0 / 1.5 | 14.6 / 14.2 | 28.7 / 35.3 | 0.3 / 0.5 | 59.0 |
| 4 | 1500 m downstream the dumps | 6.7 | 7.6 / 22.7 | 3.7 / 17.9 | 3.0 / 7.8 | 1.0 / 1.6 | 14.0 / 13.7 | 28.7 / 35.7 | 0.3 / 0.6 | 58.3 |
| 5 | Kurung River, upstream the mouth of Propadayushiy Creek | 6.7 | 9.6 / 25.7 | 3.4 / 15.0 | 4.0 / 9.3 | trace | 21.9 / 19.3 | 26.7 / 29.7 | 0.7 / 1.0 | 66.3 |
| | Natural landscapes | | | | | | | | | |
| 6 | Middle Kurung River | 7.4 | 7.8 / 27.5 | 2.8 / 16.2 | 2.0 / 6.3 | trace | 20.7 / 23.9 | 13.4 / 20.1 | 3.0 / 6.0 | 49.7 |
| 7 | Lower Elkon River | 7.8 | 16.2 / 26.8 | 6.6 / 17.9 | 3.0 / 4.3 | 1.0 / 1.0 | 51.9 / 28.2 | 27.6 / 19.0 | 3.0 / 2.8 | 109.3 |
| 8 | Aldan River, Town of Tommot | 7.4 | 20.4 / 25.8 | 8.1 / 16.9 | 6.0 / 6.6 | 1.0 / 0.7 | 86.6 / 35.8 | 23.7 / 12.5 | 2.3 / 1.7 | 148.1 |

The increased concentration of sulfates in the waters of Propadayushiy Creek in the zone of influence of the radioactive dumps is the result of the oxidation of sulfide minerals, which are satellites of uranium in an oxidizing environment. In fact, sulfuric acid of low concentration is formed in these waters, resulting in the decrease of the pH values by about 1 (collection points #2–5) as compared with the background (collection point #1). In addition, we observed an increase in water mineralization at these points by 12.7–22.5 mg/L due to the dissolution of minerals contained in weakly weathered dump rocks of the hypergenesis zone.

The radionuclides studied, in the process of water migration, differently reflect the degree and scale of technogenic pollution in the influence zone of the radioactive dumps (Table 4).

The highest concentrations of uranium and radium were detected at distances of 500 and 1000 m from the beginning of the dumps, respectively. The technogenic concentration of uranium in Propadayushiy Creek, represented by the values of the excess coefficient over the background level (11–100), significantly exceeds that of radium (4.4–9.4). When calculating the values of coefficient of excess over the background level (Kex), we used the ratio of the radionuclide content at the sample collection point in the waters of the technogenic landscape to its minimum concentration observed in the waters of natural landscapes. In this case, the following concentrations of uranium and radium were taken as background concentrations: $^{238}$U—$1.8 \times 10^{-4}$ mg/L and $^{226}$Ra—$0.5 \times 10^{-9}$ mg/L. Throughout the studied 1.5 km section of the creek and down to its mouth, we detected increased concentration of the radionuclides that exceeded their background level. Similar studies that we carried out on other rivers and creeks of the Elkon uranium region showed that the increased concentrations of radionuclides in the waters were still detected at a distance of up to 2 km from the radioactive dumps.

**Table 4.** Content of radionuclides in the surface waters of technogenic and natural landscapes of Southern Yakutia.

| # | Sampling Site | U, $n \times 10^{-4}$ mg/L | Ra, $n \times 10^{-9}$ mg/L | Ra:U |
|---|---|---|---|---|
| | | Technogenic landscapes | | |
| 1 | Propadayushiy Creek, beginning of dumps | $\dfrac{140}{78}$ | $\dfrac{2.2}{4.4}$ | $1.6 \times 10^{-7}$ |
| 2 | 500 m downstream the dumps | $\dfrac{180}{100}$ | $\dfrac{2.6}{5.2}$ | $1.4 \times 10^{-7}$ |
| 3 | 1000 m downstream the dumps | $\dfrac{93}{52}$ | $\dfrac{4.7}{9.4}$ | $5.0 \times 10^{-7}$ |
| 4 | 1500 m downstream the dumps | $\dfrac{20}{11}$ | $\dfrac{2.5}{5.0}$ | $12.5 \times 10^{-7}$ |
| | | Natural landscapes | | |
| 5 | Propadayushiy Creek, 500 m upstream the beginning of dumps | 1.8 | 2.0 | $111 \times 10^{-7}$ |
| 6 | Middle Kurung River | 3.3 | 0.7 | $21.2 \times 10^{-7}$ |
| 7 | Lower Elkon River | 2.3 | 0.6 | $26.1 \times 10^{-7}$ |
| 8 | Aldan River, Town of Tommot | 4.6 | 0.6 | $13.0 \times 10^{-7}$ |

Note: the number above the line represents the content of radionuclides, below the line is the coefficient of excess over the background level (Kex).

We also calculated the radium–uranium ratios for both radioactively contaminated waters and waters of natural landscapes (Table 4). In this case, the $^{226}$Ra: $^{238}$U ratios for background waters varied within $13–111 \times 10^{-7}$, and for radioactively contaminated waters this ratio was significantly lower and within the range of $1.4–12.5 \times 10^{-7}$. In our opinion, this interesting fact can be explained by considering the ionic composition of these waters (Table 3), increased concentration of sulfates, and the chemical features of radium as an alkaline earth metal; we assume that it migrates primarily in the form of its salts, particularly as $RaSO_4$ and $RaCO_3$. Both radium salts are insoluble in water, therefore, they precipitate where appropriate conditions for their formation are created [10].

Uranium, however, at the first stage of its water migration, will most likely be leached from the rocks as a mobile form such as soluble uranyl sulfate and soluble complex carbonates. Later, when entering the taiga–permafrost landscape, uranium is extremely prone to complexation with organic acids; migrating in ultrafresh waters of the acidic class, it forms mobile humates and, in particular, uranyl fulvates. The last statement is evident considering that surface waters in the mountain taiga zone of Yakutia contain a significant amount of dissolved organic matter that makes up 10–75% of the total amount of dissolved substances [38]. Fulvic acids are the major component of dissolved organic matter, the content of which is 5–6 times higher than that of humic acids [39].

To study the effect of organic matter on the accumulation of the radionuclides studied, silty bottom sediments enriched in the organic matter were taken in close proximity to the dumps, and the content of natural $^{238}$U and $^{226}$Ra radionuclides was analyzed. The results of the analysis showed that the content of uranium in silty deposits depends on the concentration of organic matter in them, increasing with an increase in the content of the latter. This correlation can be approximated by a logarithmic function (Figure 4). The value of the correlation coefficient between the content of $^{238}$U and organic carbon was 0.961 and statistically significant at a 0.05 significance level. As for radium-226, no statistically significant correlations were found between its concentration and sediment organic carbon. In general, these data support the results of similar studies described elsewhere [6,40].

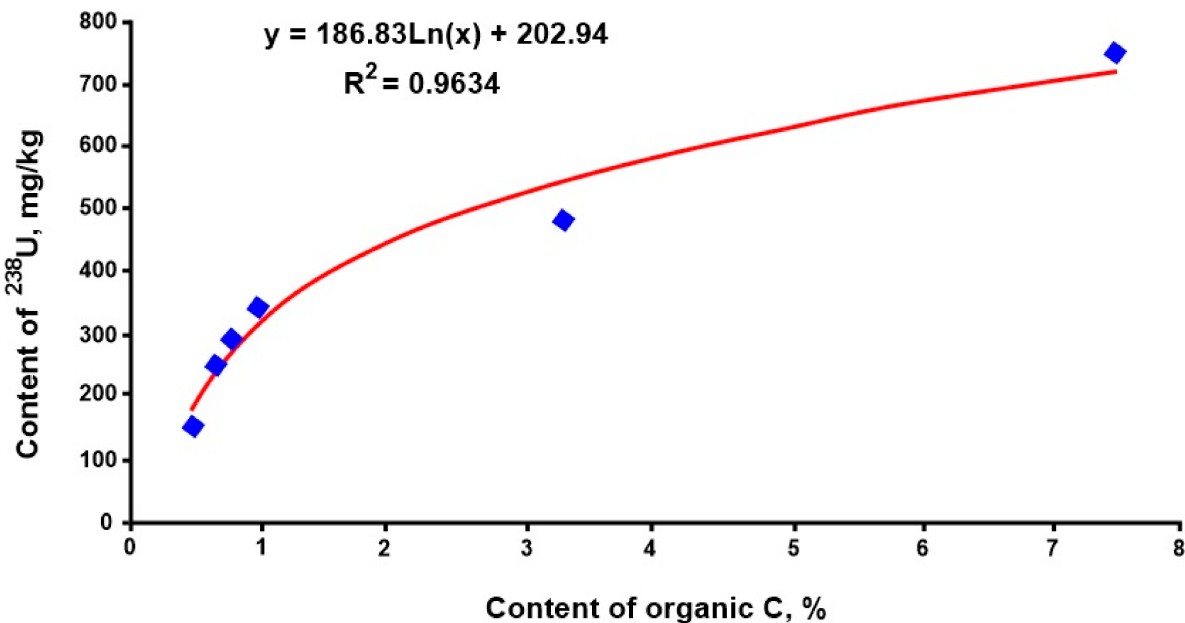

**Figure 4.** Relationship between the content of $^{238}$U (y-axis) and organic carbon (x-axis) in bottom silt deposits of Propadayushiy Creek.

### 3.3. Soil

Soil is probably the most sensitive component of a landscape, and like no other component of the ecosystem, it responds to radionuclide contamination in accordance with climatic, lithological, and geochemical features of migration of radioactive elements [41]. In conditions of radioactive contamination of the environment, soils play an important role as a major radionuclide depository in terrestrial ecosystems and as a biogeochemical barrier against their transfer to plants, animals, and a human body [42].

Here we describe the geographic and morphological characteristics of the soils studied. Soil pit 4EG-02 was dug in the lower part of the watershed slope, 80 m from dump I of the Kurung site. The geographic coordinates of the soil pit are 58°40′01.3″ N, 126°14′12.0″ E, and an absolute elevation of the area is 679.8 m above mean sea level. The morphological structure of the soil profile follows: O (0–2) -A0A1 (2–5) -A1A2 (5–8) -B (8–14) -BC (14–19) -CD (19–37) -D (37–45 cm). Soil type was determined as podzolized podbur (Entic Podzols).

Soil pit 5EG-02 was dug in the floodplain of Kurung Creek, 850 m downstream the dumps of the Kurung site. Its geographic coordinates are 58°40′12.1″ N, 126°15′06.3″ E, and an absolute elevation is 635.8 m. The profile structure of the soil follows: O (0–1) -A0A1 (1–2) -A1 (2–18) -B (18–24) -BC (24–36) -C (36–45) -(AC)⊥(45–58 cm). The soil type was classified as alluvial dark humus (Umbric Fluvisols Oxyaquic), with an active layer thickness of 58 cm.

The podzolized podbur is an automorphic soil that is formed only under conditions of atmospheric moisture, while the alluvial dark-humus soil is hydromorphic and develops on the low floodplain of Propadayushiy Creek under conditions of annual flooding. These soils also differ significantly in properties and composition (Table 5). The soil of pit 4EG-02 is characterized by a strong acidic reaction and a high content of soil organic matter in the upper organogenic horizons O and A0. Some other features of this soil include a low content of absorbed alkaline earth metals, such as $Ca^{+2}$ and $Mg^{+2}$, a high amount of exchangeable $H^+$ in the soil-absorbing complex, and sandy and sandy loam granulometric composition of the upper and the lower horizons, respectively. In the horizon BC of this soil, we observed an increased content of humus and absorbed $Ca^{+2}$, as well as a low amount of silt particles (<0.001 mm) and clay (<0.01 mm) as compared with the adjacent horizons. This is probably the result of post fire transformations within the soil after exposure to

wildfires (Table 5). However, this does not affect the content and distribution of $^{238}$U and $^{226}$Ra in this soil (Table 6).

**Table 5.** Chemical properties and physicochemical parameters of soils of technogenic landscapes of Southern Yakutia.

| Horizon | Depth, cm | pH$_{H_2O}$ | Humus, % | Exchangeable Cations, mol (eq)/kg of Soil | | | Fraction, % | |
|---|---|---|---|---|---|---|---|---|
| | | | | Ca$^{+2}$ | Mg$^{+2}$ | H$^+$ | <0.001 mm | <0.01 mm |
| *Podzolized podbur, pit 4EG-02* | | | | | | | | |
| O | 0–2 | 4.0 | 80.5 * | - | - | - | - | - |
| A0A1 | 2–5 | 4.0 | 56.2 * | 2.0 | 0.5 | 7.7 | 2.5 | 5.4 |
| A1A2 | 5–8 | 4.0 | 11.5 | 2.2 | 0.5 | 7.3 | 2.2 | 7.4 |
| B | 8–14 | 4.1 | 5.7 | 2.0 | 0.7 | 3.1 | 10.7 | 17.2 |
| BC | 14–19 | 4.2 | 6.7 | 5.0 | 1.0 | 2.8 | 6.6 | 14.4 |
| CD | 25–35 | 4.2 | 3.6 | 3.7 | 1.5 | 2.9 | 12.7 | 28.2 |
| *Alluvial dark-humus soil, pit 5EG-02* | | | | | | | | |
| A0A1 | 1–2 | 4.8 | 56.0 * | 39.2 | 13.5 | 3.9 | - | - |
| A1 | 2–18 | 4.4 | 9.9 | 8.8 | 7.8 | 2.4 | 6.4 | 13.9 |
| B | 18–24 | 4.5 | 4.8 | 4.6 | 8.3 | 1.2 | 5.2 | 10.6 |
| C | 30–40 | 5.0 | 0.6 | 3.8 | 7.5 | 0.6 | 5.3 | 10.3 |
| [AC] | 47–57 | 5.4 | 8.8 | 25.9 | 11.6 | 5.7 | 10.9 | 23.0 |

* The value of the loss during calcination is given.

**Table 6.** Content and distribution of $^{238}$U and $^{226}$Ra radionuclides in soils of technogenic landscapes of Southern Yakutia.

| Horizon | Depth, cm | $^{238}$U | | $^{226}$Ra | | $^{226}$Ra: $^{238}$U |
|---|---|---|---|---|---|---|
| | | Bq/kg | mg/kg | Bq/kg | n $\times$ 10$^{-7}$ mg/kg | |
| *Podzolized podbur, pit 4EG-02* | | | | | | |
| O | 0–1 | 85 | 6.9 | 163 | 45 | 1.92 |
| O | 1–2 | 546 | 44.3 | 652 | 180 | 1.19 |
| A0A1 | 2–5 | 190 | 15.4 | 163 | 45 | 0.86 |
| A1A2 | 5–8 | 58 | 4.7 | 50 | 14 | 0.86 |
| B | 8–14 | 44 | 3.6 | 45 | 12 | 1.02 |
| BC | 14–19 | 47 | 3.8 | 39 | 11 | 0.83 |
| CD | 19–37 | 40 | 3.3 | 43 | 12 | 1.07 |
| D | 37–42 | 15 | 1.2 | 16 | 4 | 1.07 |
| | Average * | 63 | 5.3 | 65 | 18.2 | 1.05 |
| | Kex | | 5.3 | | 4.5 | |
| *Alluvial dark-humus soil, pit 5EG-02* | | | | | | |
| O | 0–1 | 381 | 31 | 541 | 150 | 1.42 |
| A0A1 | 1–2 | 1279 | 104 | 996 | 275 | 0.78 |
| A1 | 2–4 | 1931 | 157 | 1722 | 476 | 0.89 |
| A1 | 4–6 | 2583 | 210 | 1784 | 493 | 0.69 |
| A1 | 6–8 | 725 | 59 | 455 | 126 | 0.63 |
| A1 | 8–10 | 1341 | 109 | 996 | 275 | 0.74 |
| A1 | 10–12 | 1501 | 122 | 1488 | 411 | 0.99 |
| A1 | 12–18 | 258 | 21 | 135 | 37 | 0.52 |
| B | 18–24 | 49 | 4 | 36 | 10 | 0.73 |
| BC | 24–36 | 25 | 2 | 25 | 7 | 1.00 |
| C | 36–39 | 86 | 7 | 25 | 7 | 0.29 |
| C | 39–45 | 308 | 25 | 25 | 7 | 0.08 |
| (AC) | 45–58 | 541 | 44 | 37 | 10 | 0.07 |
| | Average * | 503 | 41 | 284 | 78 | 0.56 |
| | Kex | | 13.7 | | 19.5 | |

* Weighted average content in soil profile.

The alluvial soil of the 5EG-02 soil pit is characterized by sandy loam granulometric composition, acidic reaction, and saturation of soil-absorbing complex with bases with insignificant involvement of exchangeable $H^+$. Both soils studied are also characterized by high and medium humus content in their profiles. The main feature of the alluvial soil profile is the presence of the buried horizon AC at a depth of 45–58 cm, which is characterized by an increased content of humus, exchangeable bases, and fine particles of silt and clay.

In the process of air migration, $^{238}U$ and $^{226}Ra$ contained in the radioactive dust enter the podbur surface and remain in the uppermost part of the soil profile, particularly in the forest floor or horizon O, and in humus horizon A0A1 at a depth of 2–5 cm. The weighted average contents of $^{238}U$ and $^{226}Ra$ in the profile of this soil were 5.3 mg/kg and $18.2 \times 10^{-7}$ mg/kg, and the Kex values were 5.3 and 4.5, respectively. The values of 1 mg/kg and $4 \times 10^{-7}$ mg/kg were taken as the background concentrations of these radionuclides in podburs of the study area. Based on the weighted average values of $^{238}U$ and $^{226}Ra$ specific activity, which were 63 and 65 Bq/kg, respectively, the $^{226}Ra$: $^{238}U$ ratio in this soil was 1.03, which is very close to the equilibrium state (Table 6).

The content and distribution of $^{238}U$ and $^{226}Ra$ in alluvial soil to which these radionuclides infiltrate with water were completely different from those in podbur. In contrast to podbur, where the intraprofile distribution of $^{238}U$ and $^{226}Ra$ can be described as accumulative, alluvial soil is characterized by a complex intraprofile distribution of these radionuclides. We observed two decreasing peaks of $^{238}U$ and $^{226}Ra$ content at depths of 1–6 and 8–12 cm, and the third minimal peak of $^{238}U$ content was detected at a depth of 39–58 cm. This also confirms higher mobility of $^{238}U$ as compared with $^{226}Ra$ during water migration under the natural conditions of Southern Yakutia, when uranium infiltrates the entire active layer down to the permafrost. In contrast to uranium, radium remains in the upper layer of this soil and penetrates only to a depth of 0–18 cm. The weighted average content of $^{238}U$ and $^{226}Ra$ in the alluvial dark-humus soil was 41 mg/kg and $78 \times 10^{-7}$ mg/kg, respectively, that exceeded the background levels of these radionuclides 13.7 and 19.5 times, respectively. The background concentrations of $^{238}U$ and $^{226}Ra$ in the alluvial soil studied were determined as 3 mg/kg and $4 \times 10^{-7}$ mg/kg, respectively. There was a 2.6-fold and 4.3-fold increase in $^{238}U$ and $^{226}Ra$ contamination levels, respectively, in alluvial soil, as compared with podbur, during the water migration of these radionuclides.

The weighted average $^{226}Ra$: $^{238}U$ ratio in the hydromorphic alluvial soil was 0.56, which is almost half that of the automorphic podbur, and was significantly shifted from the equilibrium state towards the excess of $^{238}U$. This is especially evident in the active layer at a depth of 39–58 cm, where the $^{226}Ra$: $^{238}U$ ratio reaches the minimum values of 0.07–0.08 (Table 6). Most likely, there are several factors and barriers contributing to the accumulation of $^{238}U$, including mechanical (permafrost), physicochemical (reducing conditions), and sorption (buried horizon); however, they do not seem to have a similar effect on $^{226}Ra$. A decrease in the radium–uranium ratio was also determined in other alluvial soils that form in the floodplains of rivers and streams at a distance of 0.9–34.0 km downstream from the radioactive rock dumps of the Kurung site. Only at 43.6 km from these dumps in the alluvial soil of the Aldan River valley does this ratio almost reach 0.96, which is very close to the equilibrium (Table 7).

**Table 7.** Content of $^{238}$U and $^{226}$Ra in alluvial soils at different distances from the rock dumps of the Kurung site in Southern Yakutia.

| River/Creek | Distance to the Dumps, km | $^{238}$U, mg/kg | $^{226}$Ra, $n \times 10^{-7}$mg/kg | $^{226}$Ra *: $^{238}$U |
|---|---|---|---|---|
| Propadayushiy Creek | 0.3 | $69.0 \pm 20.0$ ** | $157.7 \pm 23$ | 0.74 |
| Propadayushiy Creek | 0.9 | $136.1 \pm 52.0$ | $105.4 \pm 34$ | 0.22 |
| Kurung River | 2.0 | $5.1 \pm 3.0$ | $7.2 \pm 2.5$ | 0.41 |
| Kurung River | 13.0 | $7.7 \pm 2.7$ | $11.2 \pm 2.1$ | 0.42 |
| Elkon River | 34.0 | $3.8 \pm 1.2$ | $10.8 \pm 4.2$ | 0.83 |
| Aldan River | 43.6 | $2.4 \pm 0.3$ | $7.9 \pm 0.4$ | 0.96 |

* Radium in equilibrium uranium units. ** Mean value and its deviation.

Thus, we observed two types of migration of the studied radionuclides in the soils of the Elkon uranium region, namely aerotechnogenic and hydrotechnogenic, caused, respectively, by air and water dispersion of these radionuclides from contamination sources. It should also be noted that uranium and radium showed different migratory ability, which is also supported by the data obtained after the radionuclide composition analysis of plants growing in this area.

### 3.4. Plants

Of all natural heavy radionuclides, radium has the highest migration ability in the soil–plant system. Unlike uranium-238, radium-226 has no physiological barriers preventing its accumulation in plants growing on radioactively contaminated soils [40]. This statement is supported by the results of our studies on the biogenic migration of $^{238}$U and $^{226}$Ra in the technogenic landscapes of the region. In the zone influenced by the dumps, the concentration of $^{238}$U and $^{226}$Ra in trees, shrubs, and herbaceous plants is 2–80 times higher than their background levels (Table 8).

**Table 8.** Content of $^{238}$U and $^{226}$Ra in the ash of vascular plants of technogenic landscapes of Southern Yakutia, Bq/kg.

| Plant Species | Plant Organ | $^{238}$U | $^{226}$Ra | $^{226}$Ra: $^{238}$U |
|---|---|---|---|---|
| Cajander larch | Needles | 23.3 | 144.7 | 6.2 |
| (*Larix cajanderi*) | Twigs | 30.7 | 180.8 | 5.9 |
| | Trunk | 36.9 | 267.7 | 7.2 |
| Narrow-leaf willow | Leaves | 36.9 | 774.1 | 21.0 |
| (*Salix schwerinii*) | Twigs | 52.9 | 1240.8 | 23.5 |
| Siberian mountain ash | Leaves | 36.9 | 3548.9 | 96.2 |
| (*Sorbus sibiricus*) | Twigs | 49.2 | 7669.4 | 155.9 |
| | Trunk | 73.8 | 12,238.5 | 165.8 |
| Fireweed | Aerial part | 57.8 | 1,121.4 | 19.4 |
| (*Chamaeherion angustifolium*) | | | | |

The migratory capacity of $^{226}$Ra in the substrate–plant system is greater than that of $^{238}$U; therefore, the $^{226}$Ra: $^{238}$U ratio here was greater than 1.0 and shifted toward radium. This ratio was determined for different plants and their organs. For example, for the ash of different organs of cajander larch (*Larix cajanderi*), this ratio ranged from 6.2 to 7.2; for the ash of narrow-leaf willow (*Salix schwerinii*), and this ratio varied between 21.0 and 23.5; Siberian mountain ash (*Sorbus sibiricus*) was characterized by $^{226}$Ra: $^{238}$U ratio of 96.2–165.8. Compared to all of the vascular plants studied, it was Siberian mountain ash that absorbed the greatest amount of radium from the fine earth of rocks and soils.

Terrestrial and aquatic mosses also play an important role in biogenic migration of the radionuclides studied. The concentrations of $^{238}$U and $^{226}$Ra in ash samples of *Calliergon sarmentosum* and *Sphagnum teres* growing in floodplains of rivers and creeks were one to two orders of magnitude higher than those determined for *Phytidium rugorum* and *Pleurorium schreberi*, which grow in drier habitats. Moreover, the content of $^{226}$Ra in the mosses studied generally was higher by one order of magnitude, and in the case of $^{238}$U, higher by three to four orders of magnitude than that observed in trees, shrubs, and herbs (Tables 8 and 9).

**Table 9.** Concentrations of $^{238}$U and $^{226}$Ra in the ash of mosses collected in the study area, Bq/kg.

| Type of Moss by Habitat | Species | $^{238}$U | $^{226}$Ra | $^{226}$Ra: $^{238}$U |
|---|---|---|---|---|
| Terrestrial green mosses | *Rhytidium rugorum* | 762 | 350 | 0.46 |
| | *Pleurozium shreberi* | 492 | 323 | 0.66 |
| Aquatic mosses | *Sphagnum teres* | 221,400 | 8992 | 0.04 |
| | *Calliergon sarmentosum* | 263,220 | 9420 | 0.03 |

The results of special studies demonstrated that under the conditions of technogenic pollution, aquatic mosses accumulate uranium and radium primarily from the aquatic environment, whereas terrestrial mosses accumulate these radionuclides from aerial fall-out [43]. Therefore, the radium–uranium ratios in aquatic mosses are lower by one order of magnitude than those in terrestrial mosses (Table 9).

## 4. Conclusions

The main contamination source that affects soils, surface waters, and vegetation of the Elkon uranium region are the dumps of radioactive rock extracted from the depths of the Earth to the hypergenesis zone and subjected to intense physical and chemical weathering under the conditions of the cold and humid climate of Southern Yakutia. The rocks of the dumps were characterized by greatly variable initial radiation parameters, with the exposure rate values and the content of $^{238}$U and $^{226}$Ra varying within the range 1.71–16.7 μSv/h, 126–1620 mg/kg, and $428–5508 \times 10^{-7}$ mg/kg, respectively, and giving a $^{226}$Ra: $^{238}$U ratio of 1.0. As fine earth accumulates in the dumps during the weathering process, this ratio shifts from the equilibrium toward excess $^{226}$Ra and reaches 1.10–1.19, due to intensive leaching of $^{238}$U from rocks.

We observed intense water-assisted migration of $^{238}$U and $^{226}$Ra radionuclides from the dumps of radioactive rock in the technogenically disturbed landscapes of Southern Yakutia. At the same time, we observed a ten-fold decrease of the $^{226}$Ra: $^{238}$U ratio values in the waters of technogenic landscapes as compared with those of natural landscapes, due to increased uranium mobility in them in comparison with radium. The surface waters of unaffected natural landscapes were classified according to their chemical composition as of hydrocarbonate–calcium type, whereas radioactively contaminated waters were characterized mainly as sulfate–calcium. They were also characterized by decreased pH levels and increased mineralization. The maximum U and Ra contents in the waters studied were $180 \times 10^{-4}$ mg/L and $4.7 \times 10^{-9}$ mg/L, respectively, which were, respectively, 100 and 9 times higher than their background concentrations in the waters of natural landscapes. In addition, technogenic pollution of surface waters was detected at a distance of up to 2 km from the dumps during the low-water period.

We observed two types of migration of $^{238}$U and $^{226}$Ra in the soils of technogenic landscapes of Southern Yakutia, namely aerotechnogenic and hydrotechnogenic, caused, respectively, by air and water dispersion of these radionuclides from contamination sources. The intraprofile distribution of $^{238}$U and $^{226}$Ra in the automorphic podbur was characterized as accumulative, when these radionuclides remain in the uppermost part of a soil profile at a depth of 2–5 cm. In this case, the weighted average content of uranium-238 and radium-226 was 5.3 mg/kg and $18.2 \times 10^{-7}$ mg/kg, respectively; the $^{226}$Ra: $^{238}$U ratio was

1.03, and the Kex value was 5.3 and 4.5, respectively. We also observed a more complex intraprofile distribution of $^{238}$U and $^{226}$Ra in the hydromorphic alluvial soil of the study area with 2–3 distinguishable decreasing peaks of these radionuclides. The maximum increase of $^{238}$U concentration was detected in the active layer of this soil at a depth of 39–58 cm, confined to the buried soil horizon enriched with humus and fine particles of clay and silt. The weighted average content of $^{238}$U and $^{226}$Ra in the alluvial soil was 41 mg/kg and $78 \times 10^{-7}$ mg/kg, respectively, which was 13.7 and 19.5 times higher than the background concentrations of these radionuclides. The $^{226}$Ra: $^{238}$U ratio in this case was 0.56 and was shifted towards the excess of $^{238}$U. The level of radioactive contamination of this soil with uranium and radium was, respectively, 2.6 and 4.3 times higher, as compared with podbur.

During its biogenic migration in the technogenic landscapes of Southern Yakutia, $^{226}$Ra exhibits a higher mobility than $^{238}$U; therefore, the radium–uranium ratio observed in the ash of the vascular plants studied was greater than 1.0 and shifted towards the excess of $^{226}$Ra. The concentrations of $^{238}$U and $^{226}$Ra in the ash of studied plant species were 2–80 times higher than their background levels. Moreover, it was Siberian mountain ash (*Sorbus sibiricus*) that demonstrated an amazing ability to absorb radium from fine earth of soils and rocks as compared with other studied species of vascular plants. The concentrations of radionuclides studied in the ash of aquatic mosses, such as *Calliergon sarmentosum* and *Sphagnum teres*, were one to two orders of magnitude higher than those determined for terrestrial mosses *Phytidium rugorum* and *Pleurorium schreberi*. It was also found that under the conditions of technogenic pollution, aquatic mosses accumulate uranium and radium, primarily from the aquatic environment, whereas terrestrial mosses accumulate these radionuclides from aerial fallout; therefore, the radium–uranium ratios in the ash of aquatic mosses were shifted towards the excess of $^{238}$U and reached the minimum values of 0.03–0.04, common for the plants studied.

**Author Contributions:** Investigation: A.C., P.S., A.G.; Formal analysis: L.K.; Visualization: A.A. All authors have read and agreed to the published version of the manuscript.

**Funding:** This research received no external funding.

**Institutional Review Board Statement:** Not applicable.

**Informed Consent Statement:** Not applicable.

**Data Availability Statement:** The data presented in this study are available in this article.

**Conflicts of Interest:** The authors declare no conflict of interest.

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
