# Peer review of "Migration of 238U and 226Ra Radionuclides in Technogenic Permafrost Taiga Landscapes of Southern Yakutia, Russia"

_water, doi:10.3390/w13070966_

Round 1
Reviewer 1 Report
I have reviewed the article entitled "Migration of 238U and 226Ra radionuclides in technogenic perma-2 frost taiga landscapes of Southern Yakutia, Russia " . In this article, the authors measured the activity ratio of 226Ra: 238U, an explained the migration of 238U and 226Ra by the ration whether is at radioactive equilibrium (equal to 1.0) or not. The methods and analysis are commonly applied in the research of environmental radionuclides migration. Many researchers have studied the migration of natural 238U and 226Ra radionuclides in the major components of technogenic landscapes and investigated the radioecological situation at radiation hazardous objects in different natural zones However, as the authors mentioned, the data in this article were first obtained in the permafrost area in the territory of Southern Yakutia. Therefore, I suggested that the manuscript could be be accepted for publication.
Author Response
Dear Reviewer,
We want to thank you for your time and a positive decision to accept our manuscript for publication.
Best regards, Authors

Reviewer 2 Report
This is a beautiful and interesting article, the publication of which I expressly support. I therefore ask you to understand the following comments in most points not as a correction, but as a contribution to the discussion. I look at the article from the point of view of a general chemist, which is why I am not so familiar with the standard geochemical procedures and classifications.
Abstract, and Table 1 and 2:
The unit Röntgen (R) is considered to be longer outdated and therefore seems strange. Although the measuring devices used may still display Röntgen, I would suggest a conversion to the currently valid unit of Gray (respectively Nanogray).
Line 36ff:
The addition "together with liquid industrial waste" is irritating. What image do you have in mind here? Throughout the article you refer to mining dumps as a source of contamination. No picture emerges that establishes a connection with liquid anthropogenic waste. I would recommend simply deleting this addition or explaining it a little more.
Uranium paragraph; line 42ff:
In the first sentence I would suggest the following addition "occurs in relevant quantities in the earth's crust". It is debatable whether uranium or plutonium is the heaviest natural element (on earth). Plutonium is formed in very small amounts by spontaneous fission of uranium and neutron capture in uranium deposits, and Pu-244 is present in trace amounts primordially. However, the natural amounts are negligible compared to the anthropogenically produced amounts. Such a discussion can be easily circumvented by the proposed addition.
That uranium is assessed as siderophilic is surprising from a general chemical point of view. Unless this classification is self-evident from a geochemical point of view, I would suggest adding an explanatory sentence.
2 Materials and Methods
Line 127ff:
Linguistically, I would speak of measurement uncertainty rather than measurement error, since it is more a matter of statistics conditioned in the procedure than of tolerated errors in handling. Of course, this is a linguistic aspect that should be according to personal taste.
3 Results and Discussion
Table 1:
I would recommend highlighting again what was determined and what results from the assumption. Based on the methods described, I understand the values to mean that Ra-226 was determined by gamma spectrometry, while the values of U-238 result from the assumption of the equilibrium state. In my opinion, this should be explicitly mentioned again here (descriptive text and or table heading).
Line 194:
It could possibly be helpful to give a short hint at this point that the radium solubility is reduced due to the increased sulphate content through the formation of poorly soluble RaSO4 and therefore an increased leaching of uranium compared to radium is to be expected, which you then take up and explain again in section line 218ff.
Line 211:
Why is the comparison made to the minimum concentration in uncontaminated water? Since it initially seems more obvious to use the mean value as a comparison, a brief explanation would be helpful.
Line 282ff:
I fully agree with their line of argument that the radiochemical equilibrium found argues for particulate transport. However, I do not follow them as to why this necessarily takes place by means of airborne dust. (I assume this results from the terrain structure.) In my opinion, the sequence of arguments here is still insufficient and should be expanded. Why do you exclude a relevant particulate transport by means of water, e.g. as a result of snowmelt or heavy rainfall (two-dimensional occurrence of water)? At present, I see more of a contrast between dissolved and particulate transport than between hydrogenic and aerogenic transport.
Furthermore, the radiochemical equilibrium mentioned refers to the weighted average values (line 286) your argumentation, on the other hand, refers to the deposition in horizons O and A0A1, where the Ra/U ratio reaches almost 2. The values thus even contradict their argumentation.
Line 293ff:
Are the observations with Ra-226 to be seen in connection with RaSO4 as a particle, colloid or at least as a neutral particle, which just explains the observed worse or slower migration? Is it possible that dissolved Ra is not bound and therefore migrates quickly and deeply, but is also flushed out of the area under consideration?
Line 299ff:
Do you have an explanation why the alluvial soils have bound so much more compared to the Podbur? Would you rather attribute it to the higher organic content of the longer soaking?
Line 327ff:
Table values and text values differ; rounding should be considered equally in both cases:
Line 329: No, according to Table 8 the range is between 6.2 and 7.2 (not 9.0).
Line 330: According to Table 8 23.5 and not 23.4
Line 331: According to Table 8 96.2 instead of 96.1
Chapter 3.4:
You address the differences between aquatic and terrestrial mosses and show this using the Ra/U ratios. It would be very interesting if you could also explain the ratio reversal between mosses and vascular plants. Why is uranium preferentially taken up by mosses, while we preferentially take up radium in vascular plants, as explained by you? Why does the statement made for vascular plants not apply to mosses?
4 Conclusions
Line 359ff: The first sentence approximately equates U and Ra, while the differences are discussed below. This seems linguistically confusing and even contradictory. I would recommend introducing the differentiation already in the first sentence (uranium is highly leached while Ra is much less so).
Author Response
Dear Reviewer,
We appreciate your careful consideration of our manuscript, your valuable comments and suggestions, and a positive decision to accept our manuscript for publication. Following are the responses and explanations to all your comments and suggestions:
- Abstract: lines 17, 18, tables 1 and 2. Radiation exposure rate values were converted to µSv/h (according to the International System of Units (SI)).
- Line 36: The phrase: … together with liquid industrial wastes… was removed from the text.
- Line 42: The phrase… found in the earth's crust… was changed to… occurring in appropriate amounts in the earth's crust.
According to Perelman’s geochemical classification of elements (Perelman, 1989), uranium belongs to the group of lithophilic and siderophile elements, so we made the appropriate reference.
- Line 127: In our opinion, error, as well as the sensitivity of measurements, are accepted concepts from a metrological point of view.
- Table 1: In this table, 228U and 226Ra are in a state of radioactive equilibrium when the 226Ra:228U ratio equals 1. This was confirmed by different methods, including gamma-ray spectrometry, X-ray spectroscopy and radiochemical methods of uranium and radium detection, as mentioned in “Materials and Methods" section (lines 127-130).
- Lines 194, 218: We absolutely agree that under the conditions of sulfate waters of the Propadayushiy Creek, poorly soluble RaSO4 is formed.
- Line 211: We consider this approach of the determination of background uranium and radium concentrations in the waters of the study area to be quite acceptable.
- Line 282: The podbur soil pit was located on the watershed slope above the absolute height level of the radioactive dumps of the Kurung site. Here only aero-technogenic pollution was observed, when 228U and 226Ra enter the soil surface in the form of radioactive dust. The water-assisted entry of these radionuclides, as a result of melting snow or heavy rainfall, in this case, is excluded.
The shift in the radioactive equilibrium in the organogenic horizons O and A0A1, where it almost reaches 2, can be explained by the biogenic nature of radium accumulation as opposed to uranium, which is quite common in higher plants.
- Line 293: This statement is quite true for alluvial soils, since in the waters of the study area uranium migrates more intensively as compared to radium, and infiltrates to the entire depth of the soil profile, unlike radium.
- Line 299: Uranium, as opposed to radium, migrates in the surface waters of the study area mainly in the form of organomineral compounds of humic acids. This trend is supported by the function shown in Figure 4. Therefore, in alluvial soils, it accumulates in greater quantities than radium.
- Line 327, 329, 330, 331: We absolutely agree with the comments. Appropriate corrections have been made to the text of the manuscript.
- Chapter 3.4: In our opinion, mosses, as lower plants, absorb radionuclides along with other non-radioactive heavy metals, by their entire surface, while the higher plants absorb them through their root system. The roots of vascular plants have a natural barrier against uranium, as a highly phytotoxic element, while radium is selectively absorbed by these plants. This feature was very pronounced in the case of Sorbus sibiricus.
Thank you for your time!
Best regards, Authors

Reviewer 3 Report
A better selection of keywords would be recommended.
The half-life of Ra-226 (L. 54) and the other Ra-isotopes doesnot appear in the manuscript.
L. 62: "n' What is it? Please, define.
L. 66: "weight..." Which weight? Units?
L. 107: "mL" instead of "ml"
L. 126: The energy of the Cs-137 gamma-line is 661.6 keV (Iγ: 85%) and not 666 keV.
L. 128: Is not 30% error for a gamma-spectrometric measurement too high?
The authors use the CGS unit μR/h instead of μSv/h used (except USA) since 1976. Correction in whole manuscript is recommended.
Author Response
Dear Reviewer,
We appreciate your careful consideration of our manuscript, your valuable comments and suggestions, and a positive decision to accept our manuscript for publication. Following are the responses and explanations to all your comments and suggestions:
- We have rearranged the keywords in the following sequence: cryolithozone; technogenic landscapes; radionuclides 238U, 226Ra; migration; content; distribution.
- Line 54: In our opinion, there is no need to indicate the half-lives of other short-lived radium isotopes (224Ra - 3.64 days, 228Ra - 6.7 years), since the migration of these isotopes is not considered in our study. The half-life of 226Ra was added to the text of the manuscript (line 55).
- Line 62: n is the order of magnitude where the water migration rate changes Kx are equal to n – 10×n. This is directly indicated in the work of A.I. Perelman (1989) in determining the series of water migration of elements.
- Line 66: We mean the 226Ra:228U ratio equal to 3.4×10-7, when uranium and radium concentrations are expressed in percent.
- Line 107: The abbreviation of ml was changed to mL.
- Line 126: We agree that the 137Cs gamma line energy is 661.6 keV, not 666 keV. The correction has been made to the text of the manuscript.
- Line 128: We agree with the reviewer. Indeed, the error in the gamma-spectrometric determination of 228U mainly in the measured samples did not exceed 10-15%, and only in some samples with a low uranium content reached a maximum of 30%. The correction has been made to the text of the manuscript.
- Lines 17, 18, tables 1 and 2. Radiation exposure rate values were converted to µSv/h (according to the International System of Units (SI)). (This was also highlighted by other reviewers).
Thank you for your time!
Best regards, Authors
